# Towards a Hygroscopic Growth Calibration for Low-Cost PM$_{2.5}$ Sensors

Milan Y. Patel[1], Pietro F. Vannucci[1], Jinsol Kim[2], William M. Berelson[2], Ronald C. Cohen[1,3]

[1]Department of Chemistry, University of California Berkeley, Berkeley, CA 94720, United States
[2]Department of Earth Sciences, University of Southern California, Los Angeles, CA 90089, United States
[3]Department of Earth and Planetary Sciences, University of California Berkeley, Berkeley, CA 94720, United States

*Correspondence to*: Ronald C. Cohen (rccohen@berkeley.edu)

**Abstract.** Low-cost particulate matter (PM) sensors continue to grow in popularity, but issues such as aerosol size-dependent sensitivity drive the need for effective calibration schemes. Here we devise a time-evolving calibration method for the
Plantower PMS5003 PM$_{2.5}$ mass concentration measurements. We use 2 years of measurements from the Berkeley Environmental Air-quality and CO$_2$ Network sensors deployed in San Francisco and Los Angeles in our analysis. The calibration uses a hygroscopic growth correction factor derived from κ-Köhler Theory, where the calibration parameters are determined empirically using EPA AQS reference data at co-location sites during the period from 2021–2022. The parameters are found to vary cyclically through the seasons, and the seasonal cycles match changes in sulfate and elemental carbon PM
composition fractions throughout the year. In both regions, the seasonal RH dependence calibration performs better than the uncalibrated data and data calibrated with the EPA's national Plantower calibration algorithm. In the San Francisco Bay Area, the seasonal RH dependence calibration reduces the RMSE by ~40% from the uncalibrated data and maintains a mean bias much smaller than the EPA National Calibration scheme (–0.90 vs –2.73 µg/m$^3$). We also find that calibration parameters forecasted beyond those fit with the EPA reference data continue to outperform the uncalibrated data and EPA calibration data,
enabling real-time application of the calibration scheme even in the absence of reference data. While the correction greatly improves the data accuracy, non-Gaussian distribution of the residuals suggests that other processes besides hygroscopic growth can be parameterized for future improvement of this calibration.

## 1 Introduction

Particulate matter (PM) is a major air pollutant, presenting a significant human health concern. PM$_{2.5}$, particulate matter with
diameters less than 2.5 microns, has been linked with a number of health outcomes including decreased lung function, premature death, cardiovascular diseases, and cancer (Kim et al., 2015; Cohen et al., 2017). As such, local PM observations are an essential part of a system for monitoring and improving community health and wellbeing. Additionally, coincident measurements of PM and other pollutants, like CO or NO$_x$ (NO$_x \equiv$ NO + NO$_2$), can be used to elicit information on urban emissions and atmospheric processes (Fitzmaurice and Cohen, 2022). The increasing availability of low-cost PM sensors has
facilitated high-density PM monitoring and widespread use outside the scientific and professional air quality communities.

A well-documented issue with low-cost nephelometric PM sensors is their size-dependent and index of refraction dependent sensitivity. These sensors are imperfect nephelometers. They are most sensitive to sub-micron particles, and their sensitivity decreases as particles get larger with near-zero detection for particles larger than 2 microns and lower efficiency below 300nm (Kuula et al., 2020; Molina Rueda et al., 2023; Ouimette et al., 2021). For a constant particle size distribution and composition, a single scale factor can translate the observations to those made with instruments that capture the mass of the entire size distribution. However, fixed calibrations are inadequate for temporally evolving particle size and composition distributions. One of the major drivers of variability in particle size distributions is hygroscopic growth, the uptake of atmospheric water onto PM. While reference instruments such as those used in the US Environmental Protection Agency's Air Quality System (EPA AQS) measure particles under controlled, low-humidity conditions, low-cost sensors usually measure particles under ambient atmospheric conditions (Ambient Monitoring Technology Information Center, 2022; Giordano et al., 2021). Fluctuations in the relative humidity (RH) change particle size and refractive index through water uptake, both of which impact particle light scattering and subsequent detection by nephelometers (Petters and Kreidenweis, 2007; Han et al., 2020; Hänel, 1968). On longer timescales, PM size distributions and composition vary depending on primary emissions sources and secondary PM formation pathways (Mackey et al., 2021; Sayahi et al., 2019; Stavroulas et al., 2020). Theoretical calculations show that relative humidity is the largest source of uncertainty for optical particle sensors when the aerosol is hygroscopic (Hagan and Kroll, 2020). An efficient calibration scheme for nephelometric sensors must therefore account for both rapid (hourly) size fluctuations due to changes in humidity as well as long-term (monthly) variations in particle composition and hygroscopicity.

Several previous studies have reported calibrations to correct for the hygroscopic growth of particles measured with low-cost optical sensors. Crilley et al. noted the high bias of PM$_{2.5}$ mass concentrations from optical particle counters (OPCs) when the relative humidity was high and used a bias correction scheme derived from κ-Köhler Theory, which has subsequently been applied to other low-cost OPC studies (Crilley et al., 2018; Di Antonio et al., 2018). Similar bias correction schemes have also been applied to nephelometric PM sensors. Malings et al. used a hygroscopic growth correction on Plantower PMS5003 in the PurpleAir sensors as well as the Met-One NPM during a field study in Pittsburgh, PA, where separate parameters were set for summer, winter, and transition months to account for seasonal changes in the hygroscopicity of the particles based on measured speciation data (Malings et al., 2020).

Malings et al. recognized the need for seasonally variant parameterization of hygroscopic growth and implements a piecewise change in hygroscopic growth parameters to account for these seasonal changes. Here, we propose a calibration scheme for the Plantower PMS5003, a low-cost nephelometric PM sensor, whose hygroscopic growth parameters smoothly evolve through the seasons based on smooth evolution of observed composition to represent gradual changes in PM hygroscopicity over time. The Plantower is a widely-used low-cost PM sensor (Nilson et al., 2022; Molina Rueda et al., 2023; Barkjohn et al., 2021; Kumar and Sahu, 2021; Sayahi et al., 2019), so the development of regional, easy-to-implement corrections for these instruments is useful to air quality monitoring broadly.

## 2 Methods

The Berkeley Environmental Air-quality and $CO_2$ Network (BEACO$_2$N) is a high-density network of low-cost sensors spread across multiple urban centers around the globe, monitoring $CO_2$, CO, NO$_x$, O$_3$, and PM$_{2.5}$ (Shusterman et al., 2016). There are currently 57 active BEACO$_2$N sites in the San Francisco Bay Area, with additional networks in Los Angeles, CA, Providence, RI, and Glasgow, UK. Each BEACO$_2$N node enclosure contains a Plantower PMS5003 (Plantower, 2016) for PM measurements as well as an Adafruit BME280 (Adafruit Industries, 2023) for temperature, pressure, and humidity

measurements, with fans on either end of the enclosure to cycle air through the node. The Plantower PMS5003 is a nephelometric PM sensor reporting mass concentrations for PM$_1$, PM$_{2.5}$, and PM$_{10}$, though for the remainder of this work we will only discuss the PM$_{2.5}$ output measurement for this sensor. The sensor has internal calibrations, unknown to the user, that convert from scattered light intensity to PM mass concentrations. Here, we use the CF = ATM sensor output, which is recommended for outdoor PM$_{2.5}$ measurements. The calibration factor described herein is applied to this PM$_{2.5}$ mass

concentration sensor output. Plantower data was recorded every 8 seconds and averaged to hourly data points, which are used in this analysis.

    The humidity-dependent equilibrium water uptake by particles is often parametrized by the hygroscopic growth parameter, $\kappa$, which is dependent on particle composition. $\kappa$–Köhler theory can be used to derive an RH-dependent factor to account for hygroscopic particle growth (Nilson et al., 2022; Petters and Kreidenweis, 2007; Crilley et al., 2018). This can be supplemented

by an additional scaling factor, m, which can account for discrepancies between the assumed particle size distribution in the factory calibration and the true particle size distribution for the particles being measured (Hagan and Kroll, 2020; Malings et al., 2020). This leads to the following calibration algorithm:

$$PM_{2.5} = PM_{plantower} * \frac{m}{1 + \frac{\kappa}{100/RH - 1}} \; , \tag{1}$$

which has two parameters, m and $\kappa$. For a given particle, $\kappa$ can be calculated as the weighted average of the hygroscopic

growth parameters for all constituents in the particle (Petters and Kreidenweis, 2007). Similarly, one could use the weighted average of this value over a sample of particles to get their collective growth parameter. Since composition information for PM is not as widely available as total PM$_{2.5}$ mass concentration measurements, we determine $\kappa$ and m empirically and validate their values over time using seasonal trends in observed PM composition. This empirical approach is more accessible to anyone trying to implement this calibration on sensors in areas with limited speciation data. However, as a point of comparison, we

also calculate $\kappa$ using data from the EPA AQS Chemical Speciation Network. $\kappa$ values for the major aerosol components are taken from various studies in the literature (Petters and Kreidenweis, 2007; Cerully et al., 2015; Chen et al., 2022).

    The empirical parameters m and $\kappa$ in Eq. (1) are calculated using co-located BEACO$_2$N Plantower PMS5003 and EPA AQS sites, where the EPA AQS PM$_{2.5}$ provides a reference concentration for the calibration (Table 1). The fitting was performed on hourly Plantower and RH data with the Python package scipy.optimize (Virtanen et al., 2020). We describe application and

evaluation of this calibration approach to the BEACO₂N networks in the San Francisco Bay Area and Los Angeles, CA during the years 2021 and 2022. A summary of the datasets used can be found in Table S1.

In the San Francisco Bay Area, we utilize two co-location sites in the 2021–2022 period of study. These are listed in Table 1 along with a co-location site in Los Angeles, CA. Note that the Castelar ES site is located 1.06 km from EPA site 06-037-1103, whereas the two Bay Area BEACO₂N sites are on-site with their respective EPA AQS reference instruments. All three

EPA AQS sites measured hourly $PM_{2.5}$ using a Met One BAM-1020 Mass Monitor w/VSCC.

**Table 1. Co-location sites used in this study**

| BEACO₂N Site | Region | Co-Located EPA Site | Nearest AQS CSN Site |
|---|---|---|---|
| Laney | Bay Area, CA | 06-001-0012 | 06-085-0005 |
| EBMUD | Bay Area, CA | 06-001-0011 | 06-085-0005 |
| Castelar ES | Los Angeles, CA | 06-037-1103 | 06-037-1103 |

The EPA has also developed a national Plantower calibration algorithm with the following form:

$$PM_{2.5} = 0.524 * PM_{plantower} - 0.0862 * RH + 5.75 \; , \tag{2}$$

which they currently apply to PurpleAir Plantower PMS5003 measurements on their AirNow website (Barkjohn et al., 2021). This scheme, herein the "National EPA Calibration", is used throughout the paper as a point of comparison.

**3 Results and Discussion**

**3.1 RH Calibration**

Figures 1a and 1b show the calibration coefficients generated by fitting the observations from the Laney site (Table 1)

Plantower sensor to the EPA reference data using a 4-week moving window. A strong seasonal cycle is evident. The EPA AQS Chemical Speciation Network (CSN) (sites used are listed in Table 1) provides measurements for components of PM, including the major aerosol species: ammonium, nitrates, sulfates, organic carbon, and elemental carbon (EC). Figure 1c and 1d show the sulfate and elemental carbon (EC) fraction observed at the CSN site nearest to Laney. Sulfate is the most hygroscopic of the major components of aerosols, while EC is not hygroscopic (Petters and Kreidenweis, 2007; Wu et al.,

2016). For this reason, we will use these species to infer trends on the overall hygroscopicity of $PM_{2.5}$ across the seasons. The speciation data from the nearest CSN site shows strong seasonal trends where the sulfate fraction is highest in the summers while the EC fraction is highest in the winters. The fitted κ values show an appropriate response, where the particles are most sensitive to RH in the summers and least sensitive in the winters. The scaling factor, m, follows the same seasonal cycle as κ. This can be reasoned by concluding that when κ is large, particles are subject to more hygroscopic growth, and consequently

particle size distributions are shifted to larger sizes which are detected with less efficiency by the Plantower sensor. Since κ and m are calculated empirically, they may also respond to factors besides hygroscopic growth, which could account for differences seen between trends in these parameters and trends in the displayed aerosol components. Using the EPA AQS CSN data for all major aerosol components, we construct κ from the speciation data and compare it to the empirically derived κ in Figure S1. We find reasonably strong agreement in the seasonal trend for the two κ timeseries, with peaks in κ occurring

during the same times of the year.

    The calibration coefficients can be smoothed, preventing overfitting, by fitting the coefficients to a sine curve. The gray lines in Fig. 1 show the sine curve fits for 2021 and 2022, where the periods of the curves are set to 1 year. Parameters for the sinusoidal fits can be found in Table S2. Application of Eq. (1) using these smoothed κ and m parameters is herein referred to as the "Seasonal RH Dependence Calibration". This calibration is applied to the data over 2021 and 2022. Figure 2 shows a

timeseries of the pre- and post-calibration sensor data, along with the EPA AQS reference measurements. Table S3 shows that the calibrated data has a significantly improved Pearson's correlation, r, and NRMSE, especially during the winter months.

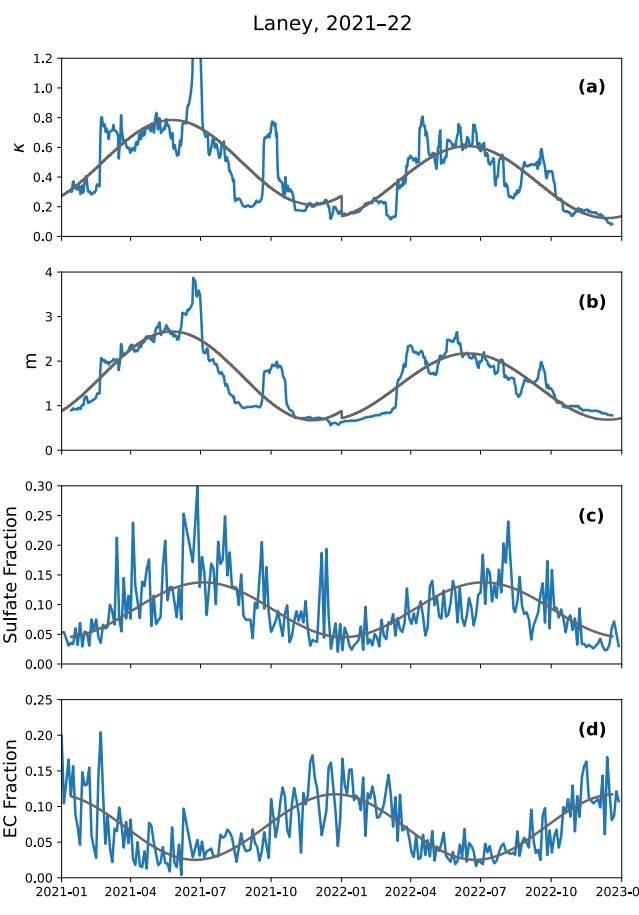

**Figure 1. (a,b) Calibration parameters generated for Laney site, and (c,d) particle speciation data from the nearest EPA AQS CSN site.**

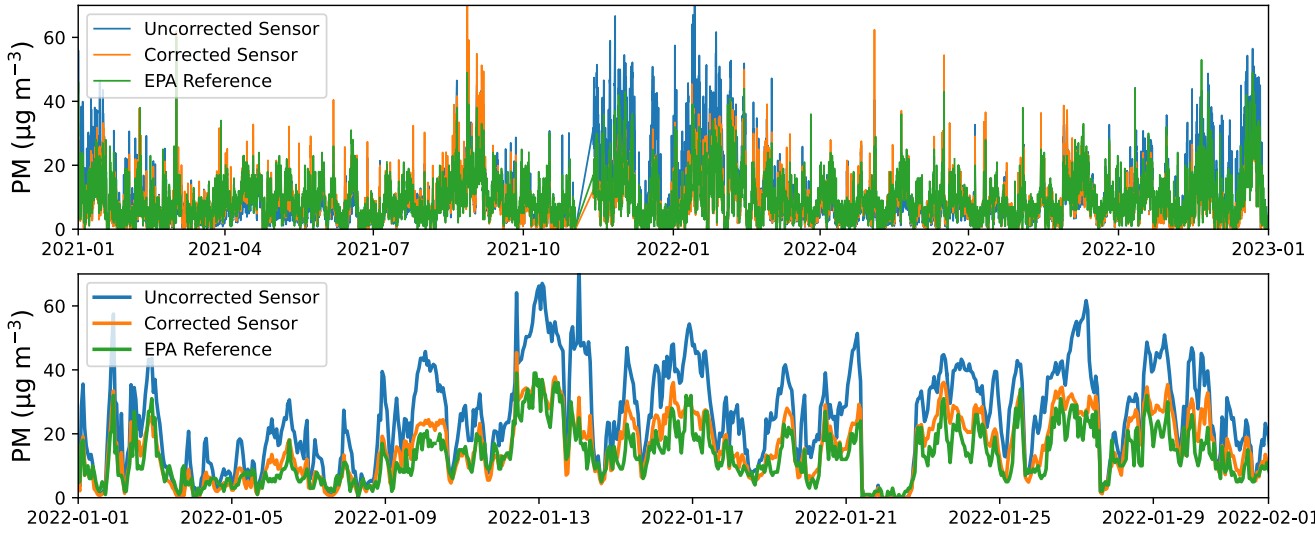


**Figure 2. Timeseries of the Laney site Plantower PM$_{2.5}$ data without and with the seasonal RH dependence calibration, compared to the co-located EPA PM$_{2.5}$ reference data for the whole two year study period (top) and for the subset of January 2022 (bottom).**

### 3.2 Evaluation

Figure 3 shows that the strong RH-dependent bias in Plantower PMS5003 outputs is removed through the implementation of

the seasonal RH dependence calibration scheme. Notably, the national EPA calibration scheme, which assumes a linear RH-dependence, does not properly account for the non-linear RH effects on particle size and detection. Figure S2 shows that the seasonal RH dependence calibration also removes the temperature dependence of the residuals. This was expected since most of the temperature dependence was likely due to covariance of temperature and RH rather than intrinsic temperature effects on sensor measurement or performance. Figure 4 shows the uncalibrated and calibrated Plantower measurements compared to

the EPA AQS reference observations for the entire two-year period. Both the National EPA calibration and the seasonal RH dependence calibration led to reductions of ~40% in the RMSE and increases of ~0.75 in the coefficient of determination (R$^2$), but the EPA national calibration introduces a large negative bias in the measurement. It is worth noting the importance of having κ and m change through time. When the calibration is applied with an optimized but constant κ and m parameter (κ = 0.311, m = 1.02), the performance of the calibration is sizably worse (R$^2$ = 0.407, RMSE = 4.884 μg m$^{-3}$, Mean Bias = -2.296

μg m$^{-3}$) than the seasonal RH dependence calibration, as apparent in Figure S3.

There were no major smoke events or other air quality events of substantial nature (e.g. multiday events, significantly high PM concentrations, etc.) during the 2021-2022 study period in the Bay Area. In August and September of 2021, there are many days with air quality warnings for smog due to winds carrying smoke from nearby fires and high temperature events worsening local pollution. If we remove data during this time period from Fig 4, the figure is virtually unchanged and the statistics are

quite similar (Fig. S4). In other time periods, we'd expect that significant air quality events could impact the analysis and should be removed from the dataset prior to fitting and applying the calibration.

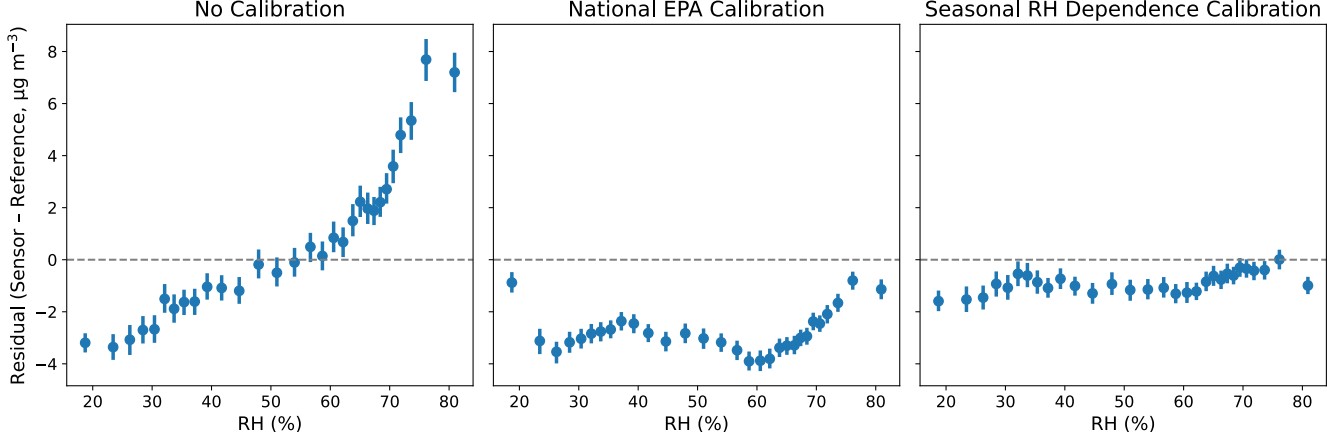

**Figure 3. Measurement residuals (sensor output – EPA AQS values) for data from the Laney site with different calibration algorithms, binned into 30 RH bins.**

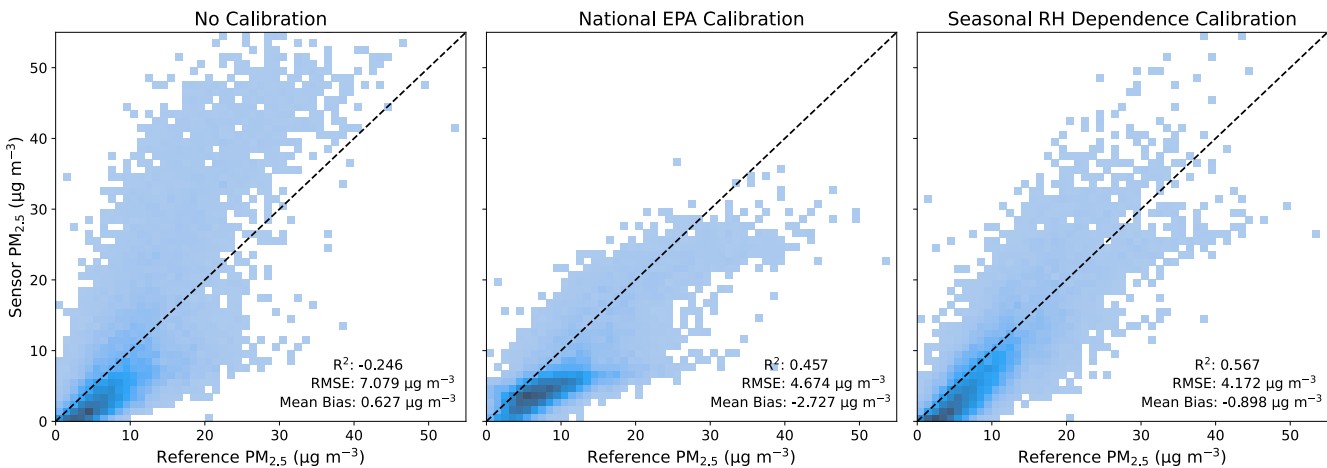

**Figure 4. Sensor predicted PM$_{2.5}$ values versus EPA reference values for Laney site from 2021–2022 with different correction algorithms. Performance metrics are the Coefficient of Determination (R$^2$), root-mean-square error (RMSE), and mean bias.**

Analyzing the distribution of errors from each of the calibration types under different mass concentration and humidity levels can help assess the completeness of each of the calibration schemes. We would expect that a complete calibration would produce zero-centered, Gaussian error distributions since all remaining errors would be from random noise in the measurement. Looking at the distribution of errors at different PM$_{2.5}$ mass concentrations (Fig. 5) it is evident that, while the seasonal RH dependence calibration produces errors more symmetric and centered near zero than the uncalibrated data and the EPA calibration data, the errors are still not perfectly Gaussian, especially when the PM$_{2.5}$ mass is high. This suggests that there are other processes and aerosol properties at play unaccounted for by this calibration, as discussed further in Section 3.5. We also explore the error distributions when κ and m are not smoothed to sine waves and find no meaningful differences between the

smoothed and unsmoothed cases when PM$_{2.5}$ ≤ 20 μg m$^{-3}$, which accounts for over 95% of the data (Fig. 5). As such, representing seasonal changes of κ and m throughout the year as sine waves is a valid approximation.

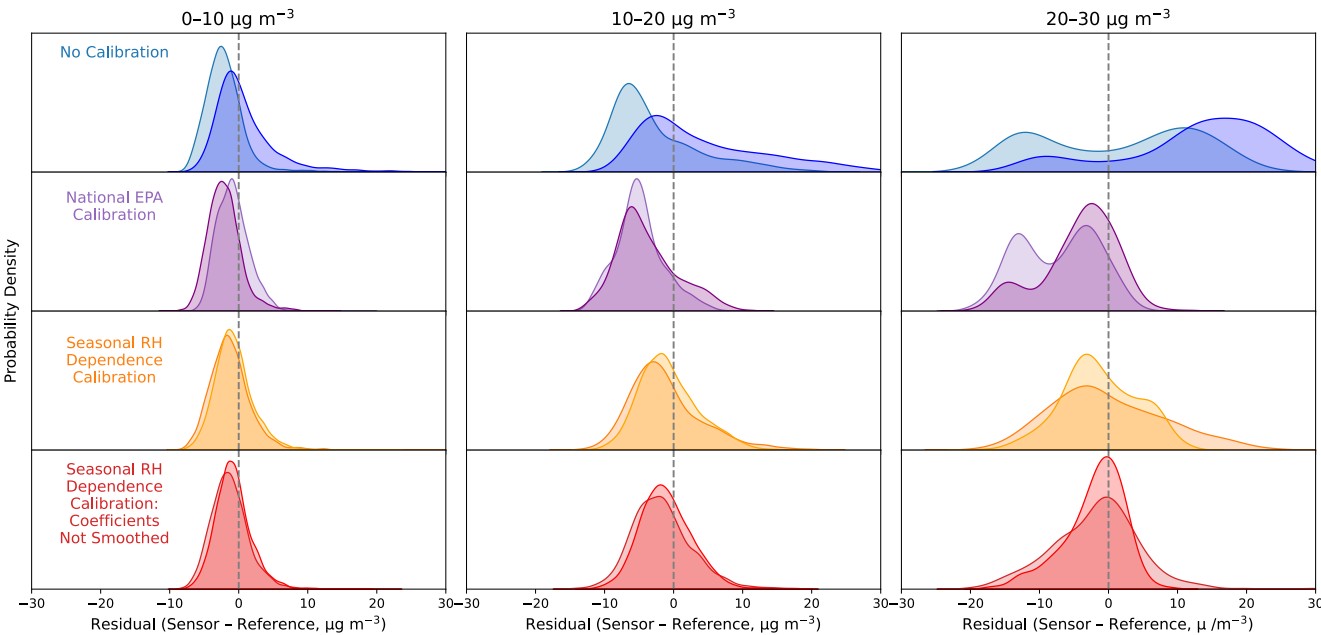

**Figure 5. Sensor residual distributions for different calibration schemes at different PM$_{2.5}$ mass concentration bins for RH < 50% (light color) and RH > 50% (dark color) conditions. The calibration scheme "Seasonal RH Dependence Calibration: Coefficients Not Smoothed" uses the calculated κ and m values as-is, without smoothing to a sine wave.**

### 3.3 Inter-Region and Intra-Region Comparisons

Given that the κ and m parameters follow trends consistent with PM speciation data, it is reasonable to assume that the parameters generated at one site can be applied to nearby sites if the particle composition is homogenous across the urban area. To further test this assumption, we independently generate calibration coefficients for another Bay Area co-location pair at the EBMUD site (Table 1). As seen in Fig. S5, the EBMUD and Laney co-location pairs independently reproduce nearly identical calibration coefficients, ensuring that the coefficients are not over-fit on one site but rather reflect regional trends in PM composition. Another concern is the possibility that individual sensors have disparate sensitivities or offsets. We find that uncalibrated measurements from 17 co-located Plantowers show strong agreement with each other (Fig. S6), with differences between sensors generally less than 1 μg m$^{-3}$, suggesting that there is little sensor-to-sensor variability in sensor performance. The seasonal RH dependence calibration scheme was tested in another urban area to ensure its generalizability beyond the Bay Area. Using a co-location site in Los Angeles, CA, we find that the aerosol composition again displays seasonality (Fig. 6), with the sulfate fraction highest in the summers and the EC fraction highest in the winters. In Los Angeles, as in the Bay Area, the trends in κ and m match the composition variations. κ and m are largest in the summers when the particles are the most hygroscopic, and smallest in the winters when the particles are the least hygroscopic. The κ sine fitting has a sizable phase

shift from the fits of the particle composition data, though this might be attributable to the low quality of the sine fitting for the LA data (Table S4). Additionally, there is poorer agreement between this empirical κ and the composition-derived κ (Fig. S7). This could indicate that in Los Angeles, factors unrelated to hygroscopicity are being captured in the empirical calculation of κ. These factors could include sub-seasonal changes in refractive index or particle size distribution, though further study is needed to provide evidence for these. Despite this, the seasonal RH dependence calibration outperforms the national EPA calibration and the uncalibrated data, with a higher coefficient of determination and lower RMSE (Table S5).

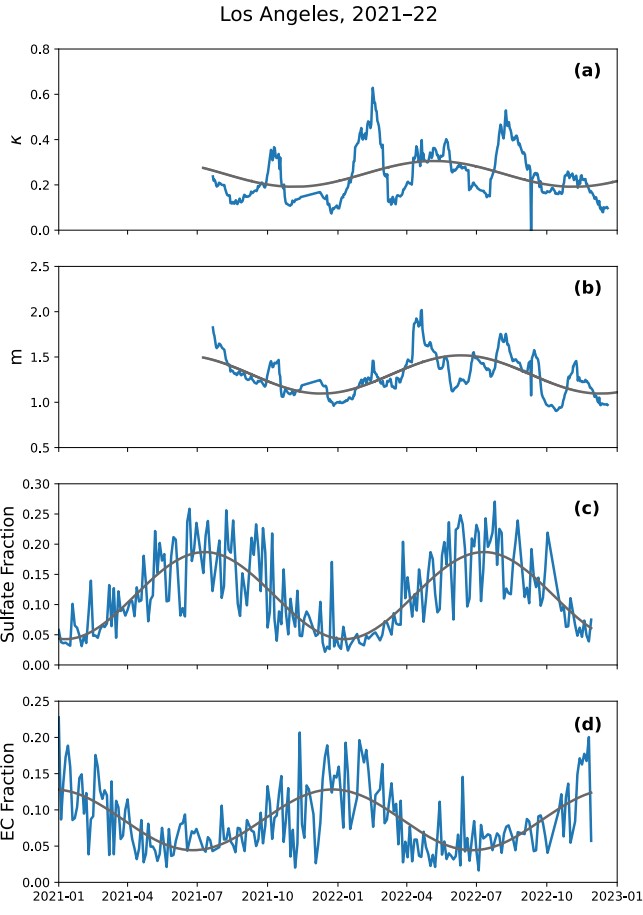

**Figure 6. (a,b) Calibration parameters generated for the Los Angeles site, and (c,d) particle speciation data from the nearest EPA AQS CSN site.**

## 3.4 Real-Time Application of the Calibration

Since the calibration coefficients generated in Fig. 1 and Fig. 6 are periodic and aerosol composition and its seasonal variation are changing slowly from year to year, it is possible to apply the seasonal RH dependence calibration to sensor measurements in real-time without the need for EPA reference measurements to be real-time as well. We test the validity of this approach by generating sinusoidal calibration coefficients in 2021 and projecting them forward 6 months into 2022 without using the 2022

reference data, as shown for data at the Laney site in Fig. 7. Like before, both the seasonal RH dependence calibration and the national EPA calibration led to significant reductions in the RMSE (~40%), and the national EPA calibration produces a significant negative bias (–2.39 µg/m$^3$, compared to 0.98 µg/m$^3$ in the seasonal RH dependence calibration). Thus, in regions with strong and stable seasonal cycles for PM composition, the sinusoidal parameters can be applied in the months following the period in which they were generated with reasonable accuracy.

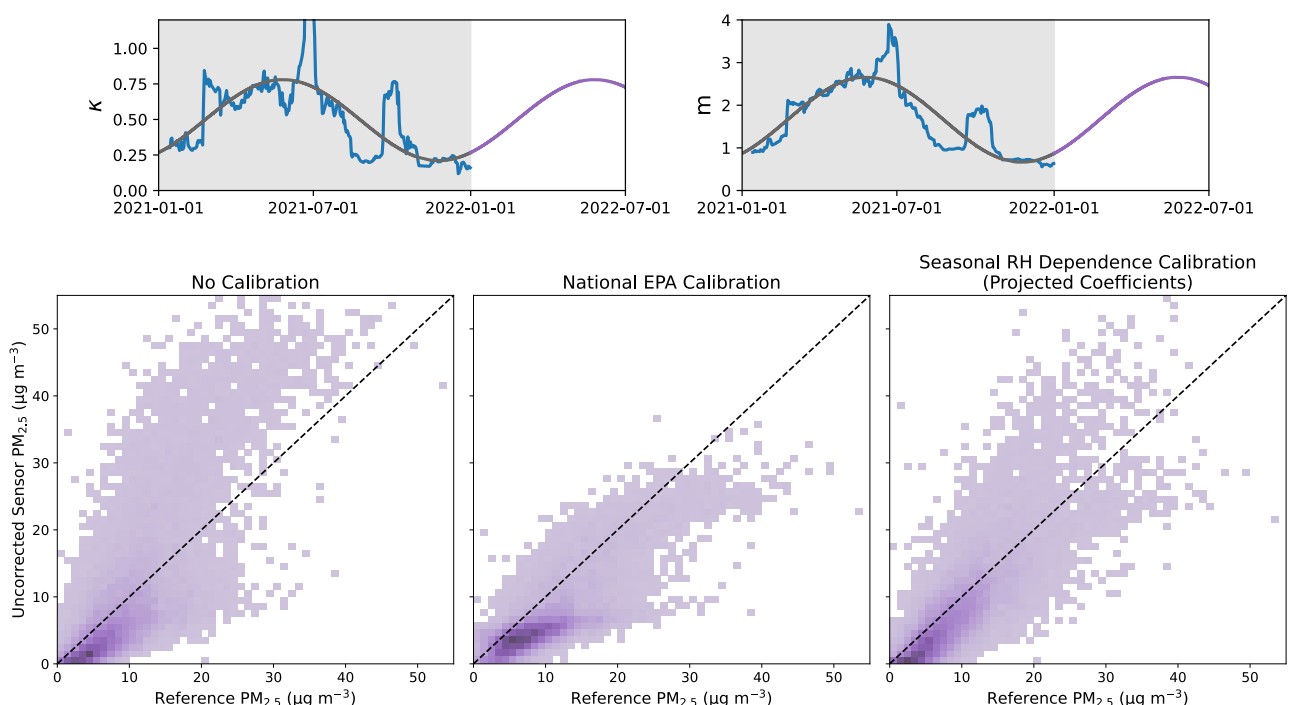


**Figure 7. (top) κ and m parameters calculated at the Laney co-location site in 2021 with sinusoidal fits projected 6 months into 2022, and (bottom) sensor predicted PM2.5 values versus co-located EPA reference values for the Laney site for Jan–Jun 2022 with different calibration algorithms.**

### 3.5 Limitations of the Calibration

There are several limitations worth noting in this calibration scheme. First, the calibration focuses on discrepancies arising from hygroscopic growth of particles, and while theoretical calculations show this to be the largest source of error (Hagan and Kroll, 2020), there are several other documented sources of error. Changes in the particle size distribution unrelated to hygroscopic growth, such as from changes in the sources of PM, are partially accounted for by the *m* parameter, but this is likely incomplete. Changes in particle composition also change the refractive index of the particles, which is ignored by this

correction scheme.

Additionally, the method assumes that particle composition is uniform across the domain and changes slowly and smoothly over the course of a year to reflect seasonal changes in particle size and composition. As such, non-seasonal changes in particle

size and hygroscopicity are not properly corrected by this method. Isolated extreme events require unique corrections, as is the case when measuring wildfire smoke or dust events (Kaur and Kelly, 2022; Holder et al., 2020). Sensors that are close to unique point sources might be consistently subject to particulate matter of a different composition and size distribution than other sensors in the network and may subsequently experience systematic biases.

There are also potential errors associated with slow drift that can occur over multiple years. Due to changes in PM sources and relative source loadings, PM composition is not expected to be the same year-over-year, so periodic recalculation of the coefficients is likely necessary. If one sensor in a network is permanently co-located with a regulatory instrument, it can be used to update the coefficients year to year for all sensors within its region. Though sensors can drift and degrade over time, current literature finds that these sensors tend to be stable for at least 3 years (deSouza et al., 2023).

**4 Conclusions**

The Plantower seasonal RH dependence calibration is aimed at addressing biases in sensor measurement caused by changes in the size distribution of PM, largely from fluctuations in relative humidity leading to hygroscopic growth and seasonal changes in particle composition that affect hygroscopicity. We provide a physically meaningful calibration scheme that is simple to define and implement for multiple regions in the United States. The seasonal RH dependence calibration utilizes only RH as an additional parameter and has calibration coefficients that reflect seasonal changes in PM speciation. This method provides a time-variant calibration scheme that can be implemented in real-time due to the periodic nature of the calibration coefficients. Speciation data provides insight and validation to the calibration parameters but is not needed in creating the calibration. Additionally, analysis of multiple co-location pairs in the Bay Area show that the calibration parameters are generally uniform within a given region, and as such this calibration can be generated for many sites using limited co-location pairs if there is reasonable confidence that aerosol speciation is uniform across the application area. Future work will be done to explore the stability of this calibration over long periods of time and work towards simple methods for correcting the unaccounted errors mentioned in the limitations section of the discussion.

The seasonal RH dependence calibration is being actively applied to the $PM_{2.5}$ measurements in the $BEACO_2N$ network, which are publicly available and can be found on their website (beacon.berkeley.edu).

**5 Code and Data Availability**

Data and code used in this paper are available at the following GitHub repository: https://github.berkeley.edu/milan-patel/Plantower-Calibration-Paper. Data from the $BEACO_2N$ network (beacon.berkeley.edu) and the EPA AQS network (epa.gov/aqs) are also publicly available online.

## 6 Author Contribution

MYP, PFV, and RCC conceptualized the work. MYP completed the formal data analysis and coding. MYP wrote the original draft, and PFV, RCC, JK, and WMB reviewed and edited the draft. RCC provided supervision for the project.

## 7 Competing Interests

Ronald C. Cohen is an associate editor of *Atmospheric Measurement Techniques*.

## 8 Acknowledgements

We would like to thank all current and former members of the BEACO2N project for their work establishing and maintaining the networks in the Bay Area and Los Angeles, CA: Alexis A. Shusterman, Virginia Teige, Kaitlyn Lieschke, Catherine Newman, Paul J. Wooldridge, Helen L. Fitzmaurice, Kevin Worthington, Naomi G. Asimow, Yishu Zhu, and Anna R. Winter.

We acknowledge the use of data from the Environmental Protection Agency's Air Quality System.

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
