# Peer review of "Towards a Hygroscopic Growth Calibration for Low-Cost PM2.5 Sensors"

_EGUsphere, 2023_

## Author Response (AR1)

**Final Author Reply to the Editor**

**CC1 (James Ouimette)**

*Comment*

Thank you for the interesting article. Could you please describe in your Methods section the source and time averaging for the RH data used in Figure 3 and your other analyses. Was it a reference RH instrument or was it the output from the PurpleAir Bosch Sensortec BME280?

Could you also describe the EPA PM2.5 reference instrument used for your PurpleAir comparison, such as MetOne BAM, Teledyne T640, GRIMM EDM Model 180, or TEOM?

*Author's Response*

Thank you for your questions! We add these details to the methods section of the revised paper.

The data is averaged to hourly data points both for the PM2.5 and the RH measurements before any calibration or analysis is performed. The RH measurements are from an Adafruit BME280 sensor which we have located next to the Plantower PMS5003 in the BEACO2N node enclosure. All three of the EPA reference sites measured hourly PM with a Met One BAM-1020 Mass Monitor w/VSCC.

*Author's Changes in the Manuscript*

The following changes have been added to the Methods section of the work:

"Each BEACO$_2$N node enclosure contains a Plantower PMS5003 (Plantower, 2016) for PM measurements as well as an Adafruit BME280 (Adafruit Industries, 2023) for temperature, pressure, and humidity measurements, with fans on either end of the enclosure to cycle air through the node."

"All three EPA AQS sites measured hourly PM$_{2.5}$ using a Met One BAM-1020 Mass Monitor w/VSCC."

**CC2 (James Ouimette)**

*Comment*

It appears from your paper that you are attempting to develop a PM2.5 calibration scheme for the Plantower PMS5003 in the PurpleAir by assuming that it is a nephelometric sensor which measures the aerosol scattering coefficient. In this case an efficient and rigorous calibration scheme would therefore involve calculating how the PMS5003 mass scattering efficiency changes with relative humidity. A recent paper (reference below) shows that the PurpleAir does not act as a perfect nephelometer. As a result of its geometry and perpendicularly polarized laser it significantly truncates the scattering signal for particles as they grow hygroscopically. Instead of the PMS5003 PM2.5 growing exponentially at high RH like an unheated nephelometer would measure, the PMS5003 PM2.5 is muted significantly at high RH. While the PMS5003 PM2.5 is still greater at high RH than a heated low-RH regulatory PM2.5 instrument such as the BAM, the PMS5003 PM2.5 is significantly lower than one would calculate by ignoring the unique attributes of the PMS5003.

You may want to consider providing the readers a better physical understanding behind the statistics you present in your paper.

Thanks again for the excellent paper.

*Ouimette, J. R., W. C. Malm, B. A. Schichtel, P. J. Sheridan, E. Andrews, J. A. Ogren, W. P. Arnott. 2022. Evaluating the PurpleAir monitor as an aerosol light scattering instrument. Atmospheric Measurement Techniques 15:655-676. doi.*

*Author's Response*

Thank you for your comment! This is a great paper, we referenced it in the original manuscript in the introduction along with several others that have studied size-dependent detection by low-cost PM sensors. We discuss in the paper that the sensitivity drops off as the particle size increases, though do not go into the technical details of this and instead leave the reader with references to dive further into the mechanisms at play. However, in light of your comment, we make it more explicit that the Plantower is not a perfect nephelometer and is therefore subject to detection issues like the ones you have pointed out.

*Author's Changes in the Manuscript*

Added to the Introduction:

Line 32: "These sensors are imperfect nephelometers and, as such, they are most sensitive to sub-micron particles, and sensitivity decays as particles get larger with near-zero detection for particles larger than 2 microns (Kuula et al., 2020; Molina Rueda et al., 2023; Ouimette et al., 2021)."

**CC3 (James Ouimette)**

*Comment*

Could you please include a brief summary of the data you used - -  the number of hours of the BAM, PMS5003, RH, etc.  Also, please include the frequency distribution of those 1-hr average data values in your Main paper or the Supplement. This will help the readers decide if your results, which are limited to urban California, are extendable to their situation.  Your paper would benefit from a discussion of how you handled the negative 1-hr ave BAM $PM_{2.5}$ values. The figure below for a typical western US town shows that 12% of the 1-hr ave BAM $PM_{2.5}$ are below zero, and that two-thirds of the values are at or below the EPA-published MDL for the BAM. How would you apply your Universal Hygroscopic Growth Calibration to their co-located PurpleAir data?  Thanks.

*Author's Response*

Thank you for your comment! We add a summary of the datasets in our revision. For the Laney site, 17024 hours of BAM, PMS5003, and RH data were used. For LA, 12701 hours of BAM, PMS5003, and RH data were used. We will also include a frequency distribution of the reference data. At the Laney site, only 0.9% of the BAM measurements are negative, and at the LA site, only 0.2% of the BAM measurements are negative. We do not remove negative values from our analysis as to not introduce any bias. For locations with very low PM concentrations, like the location you were discussing (Cheyenne WY), it might be worth applying a data selection

scheme that subsets or gives higher weight to data points at higher concentrations. However, in the cities being studied in this work, PM concentrations are higher and this issue is less of a concern.

*Author's Changes in the Manuscript*

A table of summary statistics for the datasets has been added to the SI and is referenced in the paper. Frequency distribution plots like the one pictured in the comment have been added to the SI as well.

**RC1**

*Comment*

This paper describes a new approach to calibrating low-cost PM sensors (specifically the Plantower PMS5003 nephelometer), based on a seasonal correction related to hygroscopic uptake. Sensors are colocated with more expensive PM instruments at air quality stations for 1-2 years, enabling the determination of seasonally-varying calibration coefficients using kappa-Kohler theory (eq 1). This is shown to provide a measurement of $PM_{2.5}$ mass concentration that is improved over the "raw" Plantower output (from the factory calibration) as well as over an EPA-recommended RH correction.

The results of the study seem to be sound, and the topic – calibration of low-cost PM sensors – is certainly of interest to the readership of AMT (which has published many sensor-calibration papers in the past). However a major weakness of the paper is the lack of any provided context for the work – there is virtually no discussion of previous relevant studies, including those that use kappa-Kohler theory to correct for errors in PM mass, and little discussion of how this approach can be applied and what its limitations are. These represent major shortcomings of this work, and need to be addressed before this can be considered for publication

Specific comments are listed below.

*Author's Response*

The authors thank the reviewer for their time and their comments and feedback on the manuscript.

*Author's Changes in the Manuscript*

Changes discussed in detail below, line numbers refer to lines in the original manuscript

*Comment*

1) Relationship to previous work. In describing their methods, and assessing results, the authors compare only against one other approach, the EPA Plantower correction (Barkjohn et al. 2020). But they make almost no mention of the many other studies in the literature that also correct mass concentration measurements by the Plantower (and other light-scattering-based low-cost PM sensors) for hygroscopic growth. There are at least four other studies that do this using the same general approach (kappa-Kohler theory) as the present paper: Di Antonio et al. 2018 (https://doi.org/10.3390/s18092790), Crilley et al. 2018 (https://doi.org/10.5194/amt-11-709-2018), Malings et al. 2020 (https://doi.org/10.1080/02786826.2019.1623863), and Hagan and

Kroll 2021 (https://doi.org/10.5194/amt-13-6343-2020). None of these papers are discussed; only one is cited. By not discussing these previous studies, this text inaccurately represents the state of the art of the field, as well as the novelty of the present work.

The approaches used by these (and possibly other) studies need to be discussed and compared to that of the present study, and the parts of this study that are different/novel (e.g., the sinusoidal seasonal corrections based on colocation experiments) should be highlighted. Ideally the present results would be compared against results using the approaches from these other studies (in Figs 3-4), but I recognize this might be challenging.

*Author's Response*

Thank you for your comment. The reviewer highlights the lack of contextualization of the work in the frame of current and previous literature surrounding the topic. We recognize the need add additional references in our description of the current state of knowledge surrounding low-cost PM sensor calibrations. As such, we modify the introduction to provide a brief review of the literature on this topic in our revision.

*Author's Changes in the Manuscript*

Line 44: "Theoretical calculations show that relative humidity is the largest source of uncertainty for optical particle sensors when the aerosol is hygroscopic (Hagan and Kroll, 2020)."

Line 47: "Several previous studies have reported calibrations to correct for the hygroscopic growth of particles measured with low-cost optical sensors. Crilley et al. noted the high bias of $PM_{2.5}$ mass concentrations from optical particle counters (OPCs) when the relative humidity was high and used a bias correction scheme derived from κ-Köhler Theory, which has subsequently been applied to other low-cost OPC studies (Crilley et al., 2018; Di Antonio et al., 2018). Similar bias correction schemes have also been applied to nephelometric PM sensors. Malings et al. used a hygroscopic growth correction on Plantower PMS5003 in the PurpleAir sensors as well as the Met-One NPM during a field study in Pittsburg, PA, where separate parameters were set for summer, winter, and transition months to account for seasonal changes in the hygroscopicity of the particles based on measured speciation data (Malings et al., 2020).

Malings et al. recognized the need for seasonally variant parameterization of hygroscopic growth and implements a piecewise change in hygroscopic growth parameters to account for these seasonal changes. Here, we propose a calibration scheme for the Plantower PMS5003, a low-cost nephelometric PM sensor, whose hygroscopic growth parameters smoothly evolve through the seasons based on smooth evolution of observed composition to represent gradual changes in PM hygroscopicity over time."

The following statement will replace an existing statement in the methods to better contextualize previous works:

Line 66: "This can be supplemented by an additional scaling factor, m, which can account for discrepancies between the assumed particle size distribution in the factory calibration and the true particle size distribution for the particles being measured (Malings et al., 2019; Hagan and Kroll, 2020)."

*Comment*

2) Calibration factors unrelated to hygroscopic growth. Implicit in Equation 1 and the associated discussion (lines 63-74, 98-102) is that all errors in the "raw" Plantower output are from the neglect of hygroscopic growth (and its associated change to refractive index). However this is almost certainly not the case, as many other factors can contribute. These includes issues arising from differences in the aerosols that are used in the Plantower factory calibration and those measured in the atmosphere. Key properties that may differ include size distribution (number of modes, mode median diameter, mode width, fraction of particles outside the instrument size cutoffs), refractive index, and density. These are described and/or explored in detail in earlier papers (e.g., Malings et al. 2020, Hagan and Kroll 2021), and shouldn't be simply ignored here. The authors eventually do show that other factors may be contributing to calibration error (lines 136-137), but given the previous work on the topic, these need to be mentioned much earlier in the paper. (It's also possible that some of these potential errors are seasonally varying, so can be swept into the values of m or kappa, and therefore can be corrected for by eq. 1; this should be mentioned as well.)

I think Fig 1 and Fig 6 also provide some strong evidence that other non-water-uptake factors are at play. The sulfate and EC fractions are quite similar in the two cities (Bay Area and LA, respectively) – but the kappa and m values are not. (In fact, the LA site shows higher sulfate but a much lower fitted kappa). This would suggest that the fitted kappa and m values are not entirely driven by changes in hygroscopicity, and are controlled by other factors a well; this needs to be discussed.

An alternate explanation is that the other non-sulfate, non-EC components of the aerosol (organics, nitrate…) are controlling kappa at the two sites. But this is easy to test: one just needs to compare reconstructed kappa from the speciated measurements to the fitted kappa values. (This is worth doing regardless!)

*Author's Response*

Thank you for your comment. We did not mean to imply that all issues with the Plantower output are attributable to RH dependencies/hygroscopic growth. Rather, we mean to correct the discrepancies resulting from hygroscopic growth since these are the leading error term, as noted by Hagan and Kroll (2020). With the expanded introduction to previous literature on the topic, the revised manuscript will more clearly present our method as an attempt to further improve the parameterization of hygroscopic growth which in turn produces a more reliable interpretation of Plantower output. We also briefly discussed the role of the scaling factor but have now expanded that discussion in the paper. The scaling factor, m, helps account for differences between the particle size distribution assumed by the factory calibration and the true size distribution being measured. Here we assume it changes smoothly through time and as such does not capture rapid changes in particle size distribution. Several of the other limitations of the scheme are now discussed in the new Limitations section being added in response to comment 3 below.

About non-water uptake factors, it is very likely that the empirical κ and m parameters are also responding to other factors unrelated to hygroscopic growth since the fitting is non-discriminatory. This was not discussed in the original manuscript so we will add a discussion on this to the paper.

To the point on reconstructing κ from the speciation data, this is something we had considered prior to submission of the manuscript. We had originally refrained from doing so due to ambiguities in the process including proper handling of ion pairings (EPA reports inorganic

cations and anions separately but κ values are measured and tabulated for ion pairs), uncertainty about converting organic carbon mass to organic matter mass, the generalization of organic matter (the κ value depends on the specific composition of organic matter), and biases in the absolute values of PM components reported by the EPA AQS CSN (e.g. nitrate and SVOC volatilization, particle-bound water, etc.). The precedent for doing this is mixed in the literature. Crilley et al. (2018) and Nelson et al. (2021) used an empirical approach like we do. Di Antonio et al. (2018) used a growth factor from a previous study for particles with a supposedly representative composition for urban aerosol (a mix of ammonium nitrate, ammonium sulfate, levoglucosan, succinic acid, and fluvic acid, measured as a mixture), yielding one fixed growth factor. Malings et al. (2019) compute κ from speciation data but do not discuss how they overcome the issues mentioned above. We now include a speciation-based construction of κ as a point of comparison to the empirically derived κ with the understanding that several assumptions must be made to do so, resulting in a constructed κ with large uncertainty.

*Author's Changes in the Manuscript*

As discussed in comment 1, the introduction now includes a more extended review of the prior literature on this topic.

Line 44: "Theoretical calculations show that relative humidity is the largest source of uncertainty for optical particle sensors when the aerosol is hygroscopic (Hagan and Kroll, 2020)."

Line 66: "This can be supplemented by an additional scaling factor, m, which can account for discrepancies between the assumed particle size distribution in the factory calibration and the true particle size distribution for the particles being measured (Malings et al., 2019; Hagan and Kroll, 2020)."

Section 3.5: "…the calibration focuses on discrepancies arising from hygroscopic growth of particles, and while theoretical calculations show this to be the largest source of error (Hagan and Kroll, 2020), there are several other documented sources of error…"

In regards to κ and m responding to parameters unrelated to water uptake, we now include the following discussion:

Line 102: "Since κ and m are calculated empirically, they may also respond to factors besides hygroscopic growth, which could account for differences seen between trends in these parameters and trends in the displayed aerosol components."

We now include a composition-based construction of κ in Figures S1 and S6, along with the following discussion in the main text:

Line 74: "This empirical approach is more accessible to anyone trying to implement this calibration on sensors in areas with limited speciation data. However, as a point of comparison, we also calculate κ using data from the EPA AQS Chemical Speciation Network. κ values for the major aerosol components are taken from various studies in the literature (Petters and Kreidenweis, 2007; Cerully et al., 2015; Chen et al., 2022)."

Line 102: "Since κ and m are calculated empirically, they may also respond to factors besides hygroscopic growth, which could account for differences seen between trends in these parameters and trends in the displayed aerosol components. Using the EPA AQS CSN data for all major aerosol components, we construct κ from the speciation data and compare it to the

empirically derived κ in Figure S1. We find reasonably strong agreement in the seasonal trend for the two κ timeseries, with peaks in κ occurring during the same times of the year."

Line 160: "Additionally, there is poorer agreement between this empirical κ and the composition-derived κ (Fig. S6). This could indicate that in Los Angeles, factors unrelated to hygroscopicity are being captured in the empirical calculation of κ."

*Comment*

3) Discussion of applicability and limitations of this approach. The paper describes this new approach nicely, and shows that it's an improvement over the no-correction and EPA-correction cases. But there is almost no discussion of what conditions this approach can be used for, and what its limitations are.

For example, the approach is fundamentally based on the assumption that the regional aerosol (used for calibration) provides a good description of all the aerosols to be measured. But what happens if a sensor is measuring from some local sources? (For example, if a sensor is located near a factory, or a restaurant, etc…) Or what if there is a major air quality event, such as a wildfire? If such particles are substantially different in properties than the regional aerosol particles used for calibration – not only water uptake properties but also optical properties and size distribution – then the measurements could be subject to considerable error. This error would be systematic for a given aerosol type, so could lead to consistent over- or under-estimates of certain PM classes of interest. This PM sensor accuracy issue – that it depends critically on the representativeness of the calibration aerosol – is a well-known limitation of low-cost-PM sensors, but it isn't mentioned here at all. It should be discussed, along with an examination of the sorts of errors that might arise for different aerosol types.

Another question related to applicability: just how much calibration is necessary? For example, for how long does the colocation calibration need to be done – six months, one year, or more than that? Also, over timescales of years, sensors can drift, and aerosol loadings and composition can change. Given that, for how long is the calibration accurate? It's possible the authors don't have enough data to address this at the time, but it should at least be mentioned as an important question and area for future research.

*Author's Response*

Thank you to the reviewer for this note. While we briefly identify air quality events as a limitation in the conclusion of our work, it is worth expanding and generalizing in the paper and pulling from the conclusions to the discussion. We had identified the work done by previous authors in thinking about specific calibrations for events like wildfire or dust events and marked these as a limitation to the method. In expanding on this to the reviewer's point, we will include discussion on when the assumption being made (that PM composition is uniform and slowly evolving in time) is valid and what the limitations are in maintaining this assumption. The reviewer is correct in wondering about local sources and composition. Sensors that are placed close to point sources may be subject to specific particle types with unique hygroscopicity, limiting the use of a general calibration.

To the second question of applicability, we are currently limited to about 2.5 years of data, as is presented in the study. Given the current calibration scheme, in which the coefficients are

smoothed to a sine wave with a period of 1 year, it would be best to use 1 year of data to generate the coefficients. As done in this work for 2021 and 2022, re-fitting the curves each year can help account for gradual changes in PM composition and potentially even sensor drift. We will include a discussion of this in the paper and highlight it as an area of future work.

*Author's Changes in the Manuscript*

We now include a section in the Results and Discussion to expand our talk of the limitations of the method and remove this discussion from the Conclusion section:

**" 3.5 Limitations of the Calibration**

There are several limitations worth noting in this calibration scheme. First, the calibration focuses on discrepancies arising from hygroscopic growth of particles, and while theoretical calculations show this to be the largest source of error (Hagan and Kroll, 2020), there are several other documented sources of error. Changes in the particle size distribution unrelated to hygroscopic growth, such as from changes in the sources of PM, are partially accounted for by the *m* parameter, but this is likely incomplete. Changes in particle composition also change the refractive index of the particles, which is ignored by this correction scheme.

Additionally, the method assumes that particle composition is uniform across the domain and changes slowly and smoothly over time. As such, non-seasonal changes in particle size and hygroscopicity are not properly corrected by this method. Isolated extreme events require unique corrections, as is the case when measuring wildfire smoke or dust events (Holder et al., 2020; Kaur and Kelly, 2022). Sensors that are close to unique point sources might be consistently subject to particulate matter of a different composition and size distribution than other sensors in the network and may subsequently experience systematic biases.

There are also potential errors associated with slow drift that can occur over multiple years. Due to changes in PM sources and relative source loadings, PM composition is not expected to be the same year-over-year, so periodic recalculation of the coefficients is likely necessary. Additionally, sensors can drift and degrade over time, though current literature finds that these sensors tend to be stable for at least 3 years (deSouza et al., 2023)."

*Comment*

Other comments:

1. Line 51: Malings et al. 2020 applied a seasonal kappa-Kohler-based correction, so this study is not the first to do this.

2. In Fig 1 (and Fig 6), panels c and d are provided to show that compositional seasonal trends reasonably explain the seasonal trends in kappa. However this is done purely qualitatively only. How well do the values actually correlate? (While the overall seasonal behavior matches up reasonably well, the spikes in panel c do not seem to be reflected in panel a.) Better yet, as noted above, it would be useful to reconstruct kappa based on composition measurements, and compare this value to the fitted value of kappa.

3. Equation 1: this is a reasonably well-known equation, with some version applied to low-cost PM data in previous studies – it's nearly the same as eq. 1 of Malings et al. 2020, and very

similar to eq. 4 of Crilley et al. 2018 and eq. 7 of Di Antonio 2018. A proof of it in the SI is therefore unnecessary.

4. Fig 4 (and associated discussion): were there any smoke events, or other air quality events, during this time? Where do these fall on this plot?

5. Fig 4: RMSE and biases need units.

6. Lines 137-138: another useful test would be a comparison with a non-seasonal correction (i.e., a single average value of kappa and m from the entire colocation).

7. Fig 6a: y axis should be kappa, not k.

*Author's Response*

All of these comments will be addressed/resolved in the revised manuscript. Specific changes are detailed below.

*Author's Changes in the Manuscript*

1. Line 51 was removed.

2. The difference in sampling times makes a direct comparison between the two on short timescales difficult. The speciation data is collected once every three days. The κ and m parameters are calculated using a 4-week moving window. As such, we expect the κ and m parameters to be smoother, and single measure spikes in sulfate would not be detected in them. So, comparing short-term fluctuations is not meaningful because of this. We can still more rigorously compare the seasonal trend than we had in the original manuscript. In conjunction with a comment from RC2, sinusoidal fitting parameters are now provided in the SI, which allows for quantitative comparison between the graphs. As for reconstructing κ from speciation data, please see the response above for comment 2.

3. The section of the SI deriving the equation was removed.

4. Added to Line 124: "There were no major smoke events or other air quality events of substantial nature (e.g. multiday events, significantly high PM concentrations, etc.) during the 2021-2022 study period in the Bay Area. In August and September of 2021, there are many days with air quality warnings for smog due to winds carrying smoke from nearby fires and high temperature events worsening local pollution. If we remove data during this time period from Fig 4, the figure is virtually unchanged and the statistics are quite similar (Fig. S3). In other time periods, we'd expect that significant air quality events could impact the analysis and should be removed from the dataset prior to fitting and applying the calibration."

5. Units were added to figure 4.

6. This analysis has now been performed and the results are discussed in the paper to provide the reader a better understanding for why a seasonally-changing correction is preferable. Added at Line 124: "It is worth noting the importance of having κ and m change through time. When the calibration is applied with an optimized but constant κ and m parameter (κ = 0.311, m = 1.02), the performance of the calibration is sizably worse ($R^2$ = 0.407, RMSE = 4.884 µg m$^{-3}$, Mean Bias = -2.296 µg m$^{-3}$)."

7. The y axis label in figure 7a was fixed.

**RC2**

*Comment*

The authors present a hygroscopic-growth-based correction factor for Plantower sensors and test it at a few sites in California. Overall, I found this paper to be pretty interesting, and it is well written. I recommend publication after addressing the following comments.

*Author's Response*

The authors thank the reviewer for their time and their comments and feedback.

*Author's Changes in the Manuscript*

Detailed changes are outlined below, line numbers refer to lines in the original manuscript

*Comment*

Major comments:

This can't be called a universal calibration if testing was only done at 4 sites in California. The title needs to be changed.

*Author's Response*

Thank you for this note! The title will be changed accordingly.

*Author's Changes in the Manuscript*

Title was updated to "Towards a Hygroscopic Growth Calibration for Low-Cost PM$_{2.5}$ Sensors"

*Comment*

Methods: there needs to be more detail on the methods. I didn't find certain things in the manuscript or the supplement for both empirically deriving the kappa and m coefficients and applying the correction factor. For example, how much data was used to derive the coefficients versus validate the model? How was that data selected? Was it trained on hourly or daily data? Are these monitors being calibrated just naked plantowers? Beacon nodes? What's the reference monitor? BAM?

*Author's Response*

Thank you for this note, the Methods section of this work has been expanded to included relevant details on the methodology that were originally missing as noted by this reviewer and the public commenter in CC1/CC3.

The Plantower instrument is located within the BEACON sensor enclosure alongside several other instruments. The reference instrument used at the EPA sites were Met One BAM-1020 Mass Monitors w/VSCC. The EPA reports hourly PM2.5 from these instruments. The Plantower output was recorded every 8 seconds and then averaged to hourly data. As such, the fitting was performed on hourly data. A four-week moving window was used for the fitting, and all available data was used. The data was not split into a training and validation set since the model output is not what was used for the correction. Instead, we take the modeled parameter outputs

and smooth the coefficients to a sine wave with a fixed period of 1 year. This is now described in more detail in the paper, and the exact code used in performing this analysis is publicly available in the GitHub repository for this paper.

*Author's Changes in the Manuscript*

Here are some of the changes made to the methods section meant to address these concerns and other concerns about the experimental setup:

Line 56: "Each BEACO$_2$N node enclosure contains a Plantower PMS5003 (Plantower, 2016) for PM measurements as well as an Adafruit BME280 (Adafruit Industries, 2023) for temperature, pressure, and humidity measurements, with fans on either end of the enclosure to cycle air through the node."

Line 62: "Plantower data was recorded every 8 seconds and averaged to hourly data points, which are used in this analysis."

Line 76: "The fitting was performed on hourly Plantower and RH data with the Python package scipy.optimize (Virtanen et al., 2020)."

Line 81: "All three EPA AQS sites measured hourly PM$_{2.5}$ using a Met One BAM-1020 Mass Monitor w/VSCC."

*Comment*

Figure 1 and 6: It should be possible to calculate kappa from the CSN data, instead of the sulfate and EC fractions. The authors should do this for a direct comparison rather than the qualitative comparison.

*Author's Response*

Thank you for your comment! This is something we had considered prior to submission of the manuscript. We had originally refrained from doing so due to ambiguities in the process including proper handling of ion pairings (EPA reports inorganic cations and anions separately but $\kappa$ values are measured and tabulated for ion pairs), uncertainty about converting organic carbon mass to organic matter mass, the generalization of organic matter (the $\kappa$ value depends on the specific composition of organic matter), and biases in the absolute values of PM components reported by the EPA AQS CSN (e.g. nitrate and SVOC volatilization, particle-bound water, etc.). The precedent for doing this is mixed in the literature. Crilley et al. (2018) and Nelson et al. (2021) used an empirical approach like we do. Di Antonio et al. (2018) used a growth factor from a previous study for particles with a supposedly representative composition for urban aerosol (a mix of ammonium nitrate, ammonium sulfate, levoglucosan, succinic acid, and fluvic acid, measured as a mixture), yielding one fixed growth factor. Malings et al. (2019) compute $\kappa$ from speciation data but do not discuss how they overcome the issues mentioned above. We now include a speciation-based construction of $\kappa$ as a point of comparison to the empirically derived $\kappa$ with the understanding that several assumptions must be made to do so, resulting in a constructed $\kappa$ with large uncertainty.

*Author's Changes in the Manuscript*

We now include a composition-based construction of κ in Figures S1 and S6, along with the following discussion in the main text:

Line 74: "This empirical approach is more accessible to anyone trying to implement this calibration on sensors in areas with limited speciation data. However, as a point of comparison, we also calculate κ using data from the EPA AQS Chemical Speciation Network. κ values for the major aerosol components are taken from various studies in the literature (Petters and Kreidenweis, 2007; Cerully et al., 2015; Chen et al., 2022)."

Line 102: "Since κ and m are calculated empirically, they may also respond to factors besides hygroscopic growth, which could account for differences seen between trends in these parameters and trends in the displayed aerosol components. Using the EPA AQS CSN data for all major aerosol components, we construct κ from the speciation data and compare it to the empirically derived κ in Figure S1. We find reasonably strong agreement in the seasonal trend for the two κ timeseries, with peaks in κ occurring during the same times of the year."

Line 160: "Additionally, there is poorer agreement between this empirical κ and the composition-derived κ (Fig. S6). This could indicate that in Los Angeles, factors unrelated to hygroscopicity are being captured in the empirical calculation of κ."

*Comment*

Fig 6: fitting the sine curve to the Los Angeles data seems to be a bit of a stretch here, just visually. The authors should provide details and statistics of the fit that would justify using it. The sulfate fraction and EC fraction also seem to be slightly offset from fitted Kappa at least compared to the Laney site.

*Author's Response*

Thank you for your comment. Yes, you are correct that the sine fits for the Los Angeles data are poorer than those for the Bay Area. For transparency on these fits, we will now include the fitting parameters and performance statistics to provide the reader details on these fits. The κ fit for LA is particularly bad, which could be why we see the phase shift between it and the composition data. Despite the poor fit, the corrected data with these parameters is still a sizable improvement over the uncorrected data, which we believe justifies the use of these poor fits at least as a temporary step in the progression towards a more complete calibration.

*Author's Changes in the Manuscript*

Details on the sine fits have been added to the SI and referenced in the paper.

The following has been added/modified to address the poor quality of the fits on the LA data:

Line 160: "The κ sine fitting has a sizable phase shift from the fits of the particle composition data, though this might be attributable to the low quality of the sine fitting for the LA data (Table S4). … Despite this, the seasonal RH dependence calibration outperforms the national EPA calibration and the uncalibrated data, with a higher coefficient of determination and lower RMSE (Table S5)."

*Comment*

Minor comments:

I think Malings et al 2020 AS&T did seasonally dependent RH calculations. I didn't see this one cited. I think in general there needs to be more discussion of previous work.

Line 72 methods: "Similarly, you could…". This is a bit informal and should be changed to 3rd person (one could…)

*Author's Response*

Thank you for this feedback. In conjunction with feedback from reviewer 1, I have greatly modified the introduction to include information on previous literature surrounding low-cost PM sensors, including Malings et al 2020, and where my work expands on these previous works. Additionally, I have updated Line 72 to switch to a passive voice.

*Author's Changes in the Manuscript*

Introduction greatly expanded to include background (see response to RC1 comment 1)

Changed "you could" to "one could" in the methods section

---

## Author Response (AR2)

**Final Author Reply to the Editor**

**RC1**

*Comment*

Lines 148-150: it's probably worth showing this result – the time-invariant parameterization – in a figure (probably in the supplemental), and comparing the numbers against the EPA parameterization.

*Author's Response*

We thank the reviewer for this comment and agree that such a comparison is useful. The appropriate figure will be added to the manuscript to address this.

*Author's Changes in the Manuscript*

Figure S3 was added to the supplemental and referenced in the main text. This figure shows the equivalent of Figure 4 for the version of the calibration with time-invariant parameters and compares it to the EPA and time-variant calibrations.

*Comment*

Lines 168-169 and 192-194: Potential examples of these different "physical processes" (which are probably better described as different "aerosol properties") should be given. These are mentioned later in the paper (section 3.5) but some mention here would be useful. In particular, possible reasons for these being more important in LA than in the Bay Area should be given.

*Author's Response*

We thank the reviewer for their comment. We do discuss several examples in section 3.5, though we refrain from attributing any specific reason to the LA data as we have no backing for any particular one to be true (the same is true for the non-gaussian distribution of errors the reviewer references as well). However, we can modify the manuscript to provide some speculation and point the reader to the discussion in section 3.5.

*Author's Changes in the Manuscript*

"This suggests that there are other processes and aerosol properties at play unaccounted for by this calibration, as discussed further in Section 3.5."

"This could indicate that in Los Angeles, factors unrelated to hygroscopicity are being captured in the empirical calculation of $\kappa$. These factors could include sub-seasonal changes in refractive index or particle size distribution, though further study is needed to provide evidence for these."

*Comment*

Lines 183-185: is it possible to find out if these different sensors were from the same manufacturing/calibration batch by Plantower? One could imagine that sensors calibrated together at the factory would have better agreement than ones that were calibrated separately.

*Author's Response*

We thank the reviewer for this note. In the original analysis we were looking at sensors from the same batch. We have switched the data to include sensors from two different batches for a more robust assessment of the agreement between sensors.

*Author's Changes in the Manuscript*

Figure S6 (was S5 before) has been updated with the new data and the caption is updated and explicitly states that the sensors come from two batches.

*Comment*

Lines 220-221: The text says this approach captures changes in particle composition if it "changes slowly and smoothly", but this is only true for about seasonal changes, or when there is continual calibration against regulatory grade monitors. For the main use-case of this approach – calibrating over 1-2 years then applying the calibration to deployed sensors (as in Figure 7) – year-to-year trends in particle composition (even if they are slow and smooth) are not captured. The text should be edited to reflect this.

*Author's Response*

We thank the reviewer for this comment. The use case of this approach might not have been expressed as clearly as intended. Calibrating a network of sensors in an area with this calibration scheme would involve having one sensor co-located with regulatory-grade instruments to generate coefficients, which could be updated annually to account for year-to-year trends, as is done in this work. Those coefficients could then be applied to the entire region given that particle composition is relatively uniform in that domain. This will now be made more explicit in the manuscript.

*Author's Changes in the Manuscript*

Modified Sentence: "Additionally, the method assumes that particle composition is uniform across the domain and changes slowly and smoothly over the course of a year to reflect seasonal changes in particle size and composition. As such, non-seasonal changes in particle size and hygroscopicity are not properly corrected by this method."

Modified Sentence: "There are also potential errors associated with slow drift that can occur over multiple years. Due to changes in PM sources and relative source loadings, PM composition is not expected to be the same year-over-year, so periodic recalculation of the coefficients is likely necessary. If one sensor in a network is permanently co-located with a regulatory instrument, it can be used to update the coefficients year to year for all sensors within its region. Though sensors can drift and degrade over time, current literature finds that these sensors tend to be stable for at least 3 years (deSouza et al., 2023)."

*Comment*

Figure S1 and S6: typo in legend.

*Author's Response*

We thank the reviewer for this note.

*Author's Changes in the Manuscript*

The spelling errors in the two figure legends (now figures S1 and S7) have been corrected (empircal to empirical).

**Editor Comments**

*Comment*

Pittsburgh is misspelled (missing "h" at the end) in line 54.

*Author's Response*

Thank you for this note.

*Author's Changes in the Manuscript*

The spelling error was corrected.